# Human genome-wide RNAi screen reveals host factors required for enterovirus 71 replication

Kan Xing Wu[1], Patchara Phuektes[1], Pankaj Kumar[2], Germaine Yen Lin Goh[2], Dimitri Moreau[2], Vincent Tak Kwong Chow[1], Frederic Bard[2] & Justin Jang Hann Chu[1,2]

Enterovirus 71 (EV71) is a neurotropic enterovirus without antivirals or vaccine, and its host-pathogen interactions remain poorly understood. Here we use a human genome-wide RNAi screen to identify 256 host factors involved in EV71 replication in human rhabdomyosarcoma cells. Enrichment analyses reveal overrepresentation in processes like mitotic cell cycle and transcriptional regulation. We have carried out orthogonal experiments to characterize the roles of selected factors involved in cell cycle regulation and endoplasmatic reticulum-associated degradation. We demonstrate nuclear egress of CDK6 in EV71 infected cells, and identify CDK6 and AURKB as resistance factors. NGLY1, which co-localizes with EV71 replication complexes at the endoplasmatic reticulum, supports EV71 replication. We confirm importance of these factors for EV71 replication in a human neuronal cell line and for coxsackievirus A16 infection. A small molecule inhibitor of NGLY1 reduces EV71 replication. This study provides a comprehensive map of EV71 host factors and reveals potential antiviral targets.

[1] Department of Microbiology and Immunology, National University of Singapore, Singapore 117597, Singapore. [2] Institute of Molecular and Cell Biology, Agency for Science, Technology and Research (A*STAR), Singapore 138673, Singapore. Correspondence and requests for materials should be addressed to J.J.H.C. (email: miccjh@nus.edu.sg).

Enterovirus 71 (EV71) was first isolated in 1969 from patients with disease of the central nervous system in California[1]. Since then, EV71 has re-emerged as one of the causative agents of recurring hand, foot and mouth disease (HFMD) outbreaks that affect millions worldwide[2–4]. HFMD is a mild and self-limiting febrile disease that usually affects infants and young children. However, EV71 infections are also associated with poorer disease outcomes, including aseptic meningitis and poliomyelitis-like acute flaccid paralysis, leading to death or long-term neurological sequelae[5,6]. There is currently no approved antiviral or vaccine for EV71 (refs 3,7). Given the near complete eradication of poliovirus through successful vaccination programmes, EV71 has become a medically important non-polio neurotropic enterovirus.

Enteroviruses are part of the *Picornaviridae* family of viruses, typified as non-enveloped viruses carrying a positive-sense RNA genome. With a ∼7.4 kb genome encoding just four structural and seven non-structural proteins, the successful replication of EV71 requires the participation of host factors from its initial infectious entry to eventual lytic release from its host cell. Although picornaviruses share very similar mechanisms of replication, there remain biologically significant differences that can alter their disease manifestations. Understanding the host-pathogen interactions of an infectious disease can not only inform us about its pathogenesis but can also lead to the development of host-acting broad-spectrum antivirals that are less susceptible to resistance mutations.

Research to date has uncovered some host factors involved in EV71 infection. The virus begins its infection by attaching to cell surface factors such as heparan sulfate glycosaminoglycans[8], sialylated glycans[9], annexin II (ref. 10) or PSGL1 (ref. 11). Successful entry then takes place through clathrin-mediated endocytosis via receptor SCARB2 (refs 12,13) and/or caveolin-mediated endocytosis via PSGL1 (ref. 14). The viral RNA is released from the capsid into the cytoplasm and translated non-canonically by host translation machinery, aided by internal ribosome entry site-transacting factors, such as FUBP1 (ref. 15) and hnRNPA1 (ref. 16), to produce viral proteins. These viral proteins will then evoke cell-wide changes to suppress antiviral defence and transform the cell into a virus production factory. Some of these changes include shutdown of host transcription and cap-dependent translation[17], modification of intracellular membranes to form replication complexes through coat protein (COPI) components and PI4KB (refs 18,19) and modulation of host immune responses, for example, MAVS (ref. 20) and RIG-I (ref. 21). While much of the EV71 host factor discovery work has benefited from the progress made in more established fields of poliovirus and coxsackievirus B3 (CB3) research[15,19], others have tackled the problem through mapping transcriptomic and proteomic profiles of EV71-infected cells[22,23]. Subsets of the human genome have also been screened for EV71 host factors, including siRNA libraries of endocytic and membrane trafficking genes and serine/threonine kinases[13,24]. However, the targeted and derivative strategies employed in EV71 host factor discovery thus far do not provide a comprehensive overview of the complex host-pathogen interactome during EV71 infection. Genome-wide small interfering RNA (siRNA) screens have been used to map host-pathogen interaction for several viruses. Coyne *et al.*[25] reported a comparative screen of the druggable genome library (∼5500 genes) for poliovirus and CB3 but no genome-wide screen has been reported for EV71 or any other human enterovirus to date.

In this context, we carried out an immunofluorescence-based phenotypic screen against a genome-wide siRNA library of 21,121 human genes in rhabdomyosarcoma cells (RD) to identify cellular factors that facilitate (host susceptibility factors, HSFs) or suppress (host resistance factors, HRFs) EV71 replication. Our screen identified 256 host factors involved in EV71 replication. Among these hits, we showed that the cell cycle regulators aurora kinase B (AURKB) and cyclin-dependent kinase 6 (CDK6) act as HRFs and that EV71 potentially regulates CDK6 by inducing its nuclear egress. We also found enhanced EV71 replication at specific cell cycle stages. endoplasmic reticulum (ER)-associated degradation (ERAD) components, N-glycanase 1 (NGLY1) and valosin-containing protein (VCP), were shown to be EV71 HSFs. The specific NGLY1 inhibitor Z-VAD-fmk displayed dose-dependent antiviral activities against EV71. Our findings here provide a comprehensive map of host factors involved in EV71 replication in a human cell line.

## Results

**Genome-wide RNAi screening in RD cells**. The genome-wide siRNA library consists of pools of four siRNAs per gene (siGENOME, ThermoScientific) arrayed in a series of 384-well plates. siRNA pools against *SCARB2* and *MMP20* were added to empty wells of each 384-well plate to serve as positive (EV71-inhibitory) controls while a non-targeting (NT) siRNA pool serves as the negative control (Supplementary Fig. 1). Gene knockdown was carried out by reverse-transfecting RD cells into each siRNA-containing well and incubating for 72 h, after which, infection with a clinical isolate of EV71 (5865/SIN/000009) was carried out at a multiplicity of infection (MOI) of 1 for 12 h. The infected cells were then fixed, immunostained for EV71 structural protein, VP0/VP2, and counterstained for nuclei. Images were captured by automated microscopy and image analyses gave an infection rate based on the number of viral antigen-expressing cells divided by the number of nuclei (proxy of total cell number) (Fig. 1a). The genome library was screened in three independent replicates to ensure reproducibility. We observed good correlation between the replicate screens with Pearson correlation coefficients of greater than 0.50 between independent sets (0.716, 0.552 and 0.571) (Supplementary Fig. 1c). Consistent performance for the internal controls can also be observed from the tight distribution of Z-scores for each well type from all three replicate screens (Supplementary Fig. 1d and e). To normalize the infection rate across various plates, the Z-score per gene was calculated against the mean and standard deviation of all samples per plate. The average Z-score for infection rate ($Z_{av}$) of three independent screens was then used to identify HSFs and HRFs based on robust cutoffs of $Z_{av} < -1.73$ ($Z_{av} < Z_{SCARB2} + 2SD_{SCARB2}$) and $Z_{av} > 1.86$ ($Z_{av} > Z_{NT} + 2SD_{NT}$) respectively (Supplementary Fig. 1f and g). Toxic siRNA knockdowns were removed based on the reduction of total cell number across three independent screens ($Z_{Nuclei} < -2$).

Based on these determined thresholds, we identified 905 host factors involved in EV71 replication, of which 190 are HRFs and 715 are HSFs. It is reassuring to note that the top two HSFs identified from our primary screen were *SCARB2* and *COPB2*, an essential COPI coatomer component. In addition, our screen also picked up other reported EV71 host factors, like *MINK1, PI4KB* and *FUBP1*, as among our top hits (Supplementary Data 1)[24]. Gene ontology analysis of the screening hits identified 635 genes and categorized them into 14 different biological processes, where more than half are involved in metabolic (25.5%) and/or cellular processes (21.2%). Grouping these hits by protein class revealed nucleic acid binding ($n = 103$), receptor ($n = 71$) and transcription factor ($n = 56$) as the top three classes of proteins identified in the screen (Supplementary Fig. 2).

Pathway enrichment analyses (Reactome, KEGG and gene ontology) identified 277 of the screening hits to be in interacting pathways ($P < 0.005$, Hypergeometric test) (Fig. 1b). Expectedly,

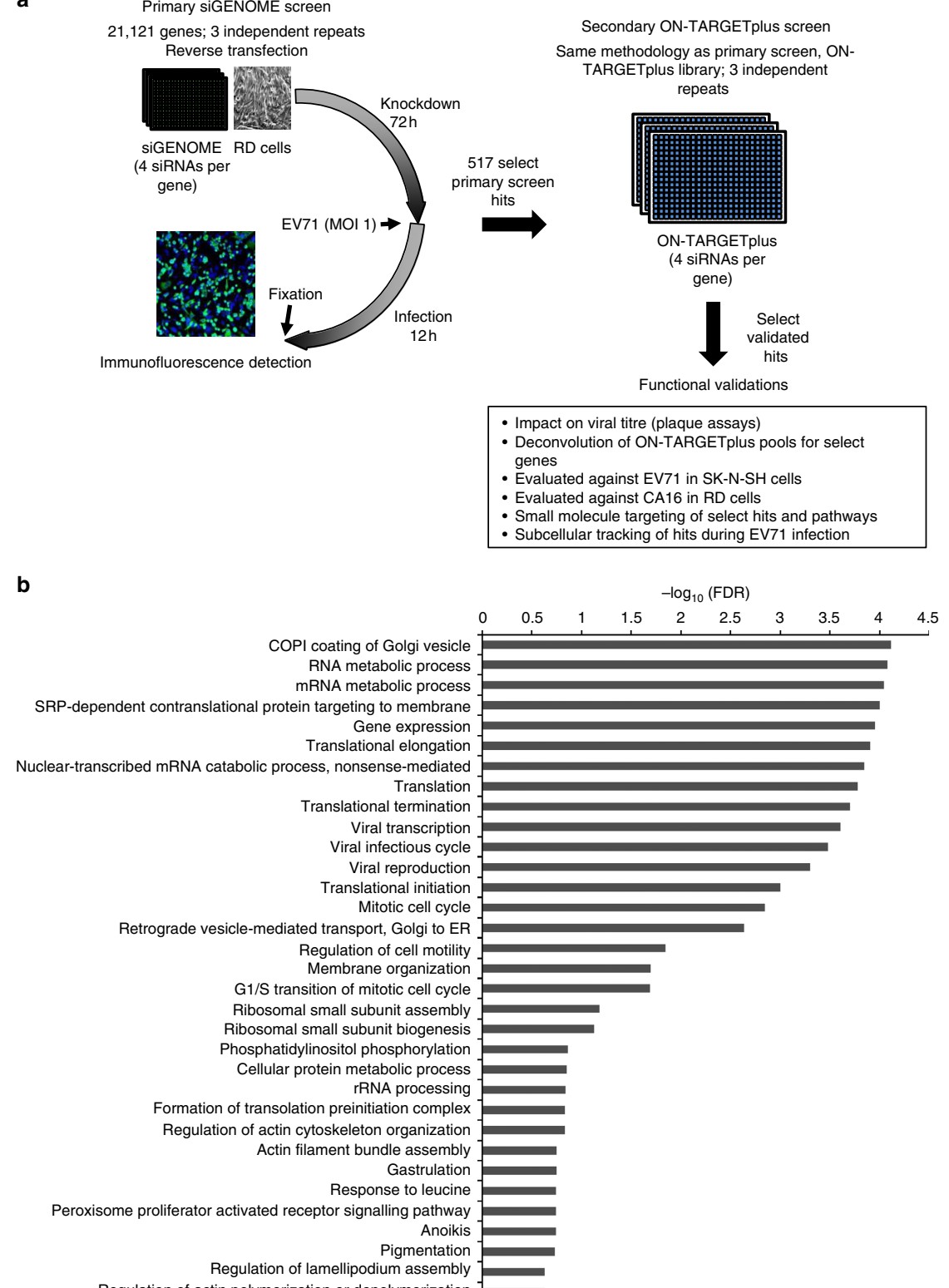

**Figure 1 | Genome-wide siRNA screen setup and functional classification of screening hits.** (**a**) Graphical summary of the study workflow from genome-wide screen to eventual functional validation of select hits. (**b**) Classification of screening hits using Gene Ontology (GO) database revealed overrepresented pathways, shown here as the log of the given false discovery rate (FDR) with all shown pathways having $P < 0.005$ (Hypergeometric test).

we found an enrichment of genes from known viral replication processes such as translation (ribosomal genes and eukaryotic initiation factors) and membrane trafficking (COPI complex, *ARCN1, PI4KB*). Other known EV71 host pathways identified in our screen include phosphatidylinositol-mediated signalling (*PIK3C3* and *PIK3CG*), clathrin-mediated endocytosis (*AP2A1* and *BLOC1S1*) and heparan sulfate biosynthesis (*HPSE* and *EXT2*)[26] (Fig. 1b). PSGL1 and caveolar components were not

identified as hits in our screen. This could be due to insufficient knockdown of these factors at screening conditions or may reflect the functional differences between EV71's entry pathways. Receptor-entry studies have revealed SCARB2 to be the 'complete' receptor that can facilitate EV71 attachment, uncoating and entry via clathrin-mediated endocytosis. In contrast, entry via PSGL1/caveolin-mediated endocytosis resulted in a relatively lower viral replication rate due to PSGL1's inability to trigger virus uncoating[27].

Besides the identification of known EV71 host factors and general viral replication pathways, the enrichment analyses also revealed cellular processes not typically associated with picornaviral infections. Genes involved in the control of mitotic cell cycle were also enriched in the screening hits, with most acting as EV71 HRFs. We then carried out a series of validation assays to confirm and characterize some of these novel EV71 host factors.

**Validation of select hits with distinct siRNA pools.** Of the 905 hits obtained from the siGENOME-based library screening, we selected 517 genes for validation using chemically modified ON-TARGETplus siRNA pools that have sequences distinct from those used in the primary screen. This was done to minimize false positives due to off-target effects. The genes were selected for their novelty as host factors for EV71. We also filtered out non-protein-coding genes and predicted or under characterized genes due to limitations in investigating them efficiently at this stage. The secondary validation was carried out in three independent screens using the same method as the primary screen. However, infection rate here was normalized against the control siRNA pools for SCARB2 and NT in each plate before HRFs or HSFs were determined (Supplementary Fig. 3). We validated 257 out of the 517 genes using a different pool of siRNAs to give a validation rate of 49.7%, which is comparable to similar studies[25].

**Functional validations.** To further narrow down the candidates for downstream characterizations, we carried out a series of functional validations on select hits. We first tested 48 of the validated screening hits in plaque reduction assays to assess the impacts of their knockdown on infectious virus production (Supplementary Fig. 4). siRNA pools of 16 genes (4 HSFs and 12 HRFs) that showed relatively greater effect on viral titre were then selected for siRNA pool deconvolution (Supplementary Fig. 4a). To rule out cell-line-specific effects and address the neurotropism of EV71, the single most effective siRNA for each of these 16 genes was tested against EV71 in a human neuronal cell-line, SK-N-SH. The siRNA knockdown of all 16 genes in SK-N-SH was found to affect EV71 infection in a similar fashion as that observed in RD cells (Fig. 2). These results suggest a conservation of the tested host factors in neuronal cells and rule out cell-line-specific effects. We also found no cell-line-dependent differences in knockdown efficiencies of siRNAs used against select hits in RD and SK-N-SH cells (Supplementary Fig. 7a). Infection endpoint for experiments using SK-N-SH cells were taken later at 24 h.p.i. as compared to RD cells (12 h.p.i.) in order to reflect the slower replication kinetics of EV71 in SK-N-SH cells (Supplementary Fig. 7b). In addition, we evaluated the same set of genes against another aetiological agent of HFMD, coxsackievirus A16 (CA16), in RD cells. The siRNA knockdown of each of the 16 genes resulted in similar inhibition or promotion of CA16 infection as observed for EV71 (Fig. 2a). CA16 is phylogenetically closest to EV71 among the human enteroviruses. The conservation of the 16 host factors shown here likely reflects this close relationship. Considering these functional validation data along with pathway analyses and the existing literature for EV71 and

picornaviruses, we selected a few genes for characterizations in orthogonal studies using small molecule inhibitors, overexpression of wild-type and/or loss-of-function mutants and immunofluorescence tracking.

**Cell proliferation genes control replication of EV71.** Consistent with the pathway enrichment analyses of our primary screening hits, we continued to pick up several proliferation and cell cycle control genes as top hits throughout our validation screens. Among the 16 genes evaluated, five are involved in cell proliferation or growth factor signalling pathways (CDK6, AURKB, CIZ1, DTX1, TGFβ1I1; Fig. 2).

CDK6, aurora kinase B (AURKB) and Cip1-interacting zinc finger protein 1 (CIZ1) are directly implicated in cell cycle regulation. CDK6 typically functions in driving G1 progression and controls subsequent G1/S phase transition through its phosphorylation of retinoblastoma protein (pRB) (ref. 28). AURKB regulates G2/M phase progression and mitotic exit at the spindle assembly checkpoint by sensing and driving proper kinetochore assembly for chromosomal segregation[29]. CIZ1 has been characterized as a p21$^{Cip1}$ interacting protein that may play a role in regulating DNA replication[30]. CIZ1 knockdown has also been reported to inhibit cell proliferation and increase the number of cells in G0/G1 phase while decreasing cells in S phase[31]. While depletion of CIZ1 resulted in inhibition of EV71 replication in RD and SK-N-SH, the knockdown of CDK6 and AURKB resulted in increased EV71 replication in both RD and SK-N-SH cells (Fig. 2). Similarly, treatments with small molecule inhibitors targeting kinase activities of CDK6 (PD00332291) and AURKB (AZD1152-HQPA) gave dose-dependent increases in EV71-infected RD cells that are comparable to the siRNA depletion studies (Fig. 3).

In order to confirm the roles of CDK6 and AURKB in restricting EV71 infection, we generated populations of RD cells stably overexpressing CDK6, AURKB and their kinase-inactive mutants. CDK6-overexpressing cells could still support EV71 infection but at a significantly lower percentage of 40% ($P < 0.001$, Student's T-test) as compared to vector-only expressing cells, while the kinase-inactive mutant, D166N (CDK6 DN), did not result in much reduction (Fig. 3b). These results suggest that CDK6 activity can restrict EV71 replication and corroborated the observations from siRNA knockdown and small molecule inhibition of CDK6. CDK6's functions in cell cycle and transcription control require its nuclear localization[32]. We tracked the cellular localization of endogenous CDK6 upon EV71 infection using immunofluorescence and found a cytoplasmic redistribution of nuclear CDK6 during late-stage EV71 infection (Fig. 3c). These observations may represent a mechanism in which EV71 regulates an HRF to promote its own infection. Overexpression of AURKB, however, did not result in any negative impact on EV71 replication. Instead, the overexpression of the kinase-dead mutant (K106R, AURKB DN) resulted in an 80% decrease ($P < 0.001$, Student's T-test) of EV71-infected cells (Fig. 3e).

In order to observe the impacts of these overexpressions on the cell cycle, we performed flow cytometric analyses on propidium iodide-stained cell populations. The stable cell lines overexpressing CDK6, AURKB or their dominant negative counterparts were compared against the empty vector expressing cell line (V5) (Supplementary Fig. 5). All samples were serum starved for 16 h before reintroduction to full serum media and analysed at every 4 h. As can be seen in Supplementary Fig. 5b, the overexpression of AURKB did not result in significant perturbations in proportion of cells in G0/G1, S or G2/M phases relative to V5-expressing cell line. In contrast, the overexpression of

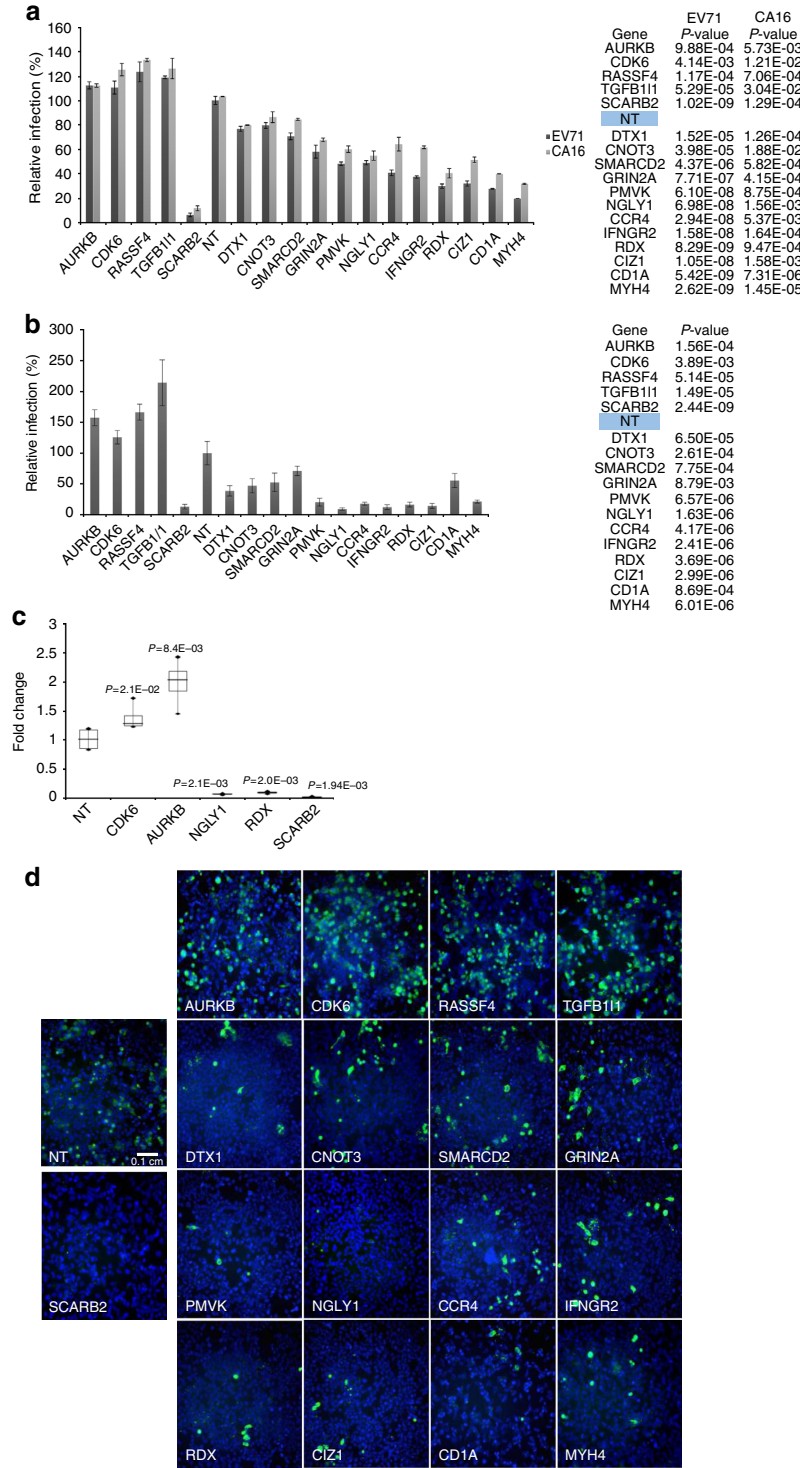

**Figure 2 | Evaluation of select hits against EV71 or CA16 in RD cells and EV71 in the neuronal cell line SK-N-SH. (a)** RD cells were infected with EV71 or CA16 at MOI 1 before relative infection rates were determined at 12 h.p.i. based on immunofluorescence quantification and normalized against NT controls. **(b)** SK-N-SH cells were infected with EV71 at MOI 1 and evaluated at 24 h.p.i. using immunofluorescence-based quantification of infection rates normalized against NT controls. **(c)** qRT-PCR detection of viral RNA at 12 h.p.i. showing the effects of siRNA knockdown of CDK6, AURKB, NGLY1 or RDX in RD cells on EV71 replication. Fold changes relative to NT-treated cells were calculated and results show box representing the upper quartile, median and lower quartile from three independent experiments while bars indicate highest and lowest reading for each sample. *P*-values are shown for two-tailed, Student's *T*-test performed against NT controls. **(d)** Representative images of the effect of gene knockdown on EV71 replication in SK-N-SH are shown, with the green signal staining for VP0/VP2 and the blue for nucleus. Representative scale bar for all micrographs in this panel can be found in the NT micrograph with an indicated length of 0.1 cm. **(a)** and **(b)** show means of results from three independent experiments and error bars represent standard deviation. *P*-values shown are for one-tailed, Student's *T*-test performed for each sample against NT controls.

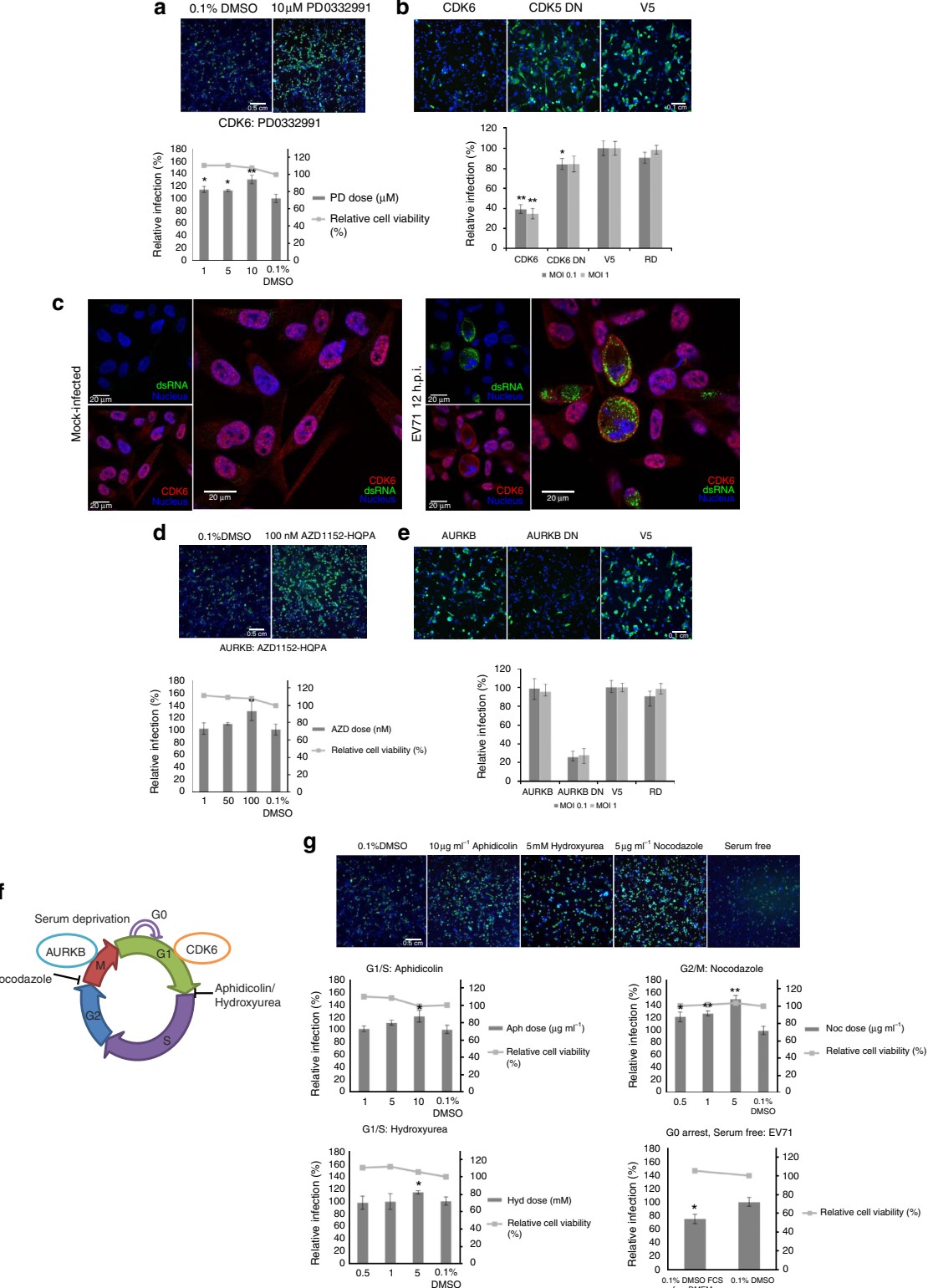

**Figure 3 | CDK6 and AURKB restrict EV71 infection.** (**a**) Dose-dependent inhibition of CDK6 using PD00332291 or (**d**) AURKB using AZD1152-HQPA, for 16 h before infection and throughout infection resulted in increased rate of EV71 infection. Bars represent mean infection rate relative to 0.1% DMSO and error bars represent standard deviation. (**b**) EV71 infection in cells stably overexpressing CDK6 or its kinase inactive mutant CDK6 DN or (**e**) AURKB and its kinase inactive mutant AURKB DN. Bars represent mean infection rate relative to V5 (empty vector) and error bars represent standard deviation. (**c**) Immunofluorescence tracking of CDK6 in mock-infected and EV71-infected RD cells at 12 h.p.i.. Red signal indicates CDK6, green for dsRNA and blue for nuclei. (**f**) Diagram showing the relative functional roles of AURKB, CDK6 and inhibitors used in the mitotic cell cycle. (**g**) Cell cycle inhibition of RD cells at specific stages enhances EV71 replication. All inhibitor studies here involved a 16 h pre-infection treatment followed by continued treatment after infection-adsorption until 12 h.p.i.. Bars represent mean relative infection rates and error bars show standard deviations. All results shown are of two-independent experiments. Two-tailed, Student's T-test was performed against controls, *P < 0.05 and **P < 0.001. Representative scale bars for all micrographs in (**a,d,g**) can be found in the control 0.1% DMSO micrograph with an indicated length of 0.5 cm. The representative scale bars for (**b,e**) can be found in the control V5 micrograph with an indicated length of 0.1 cm.

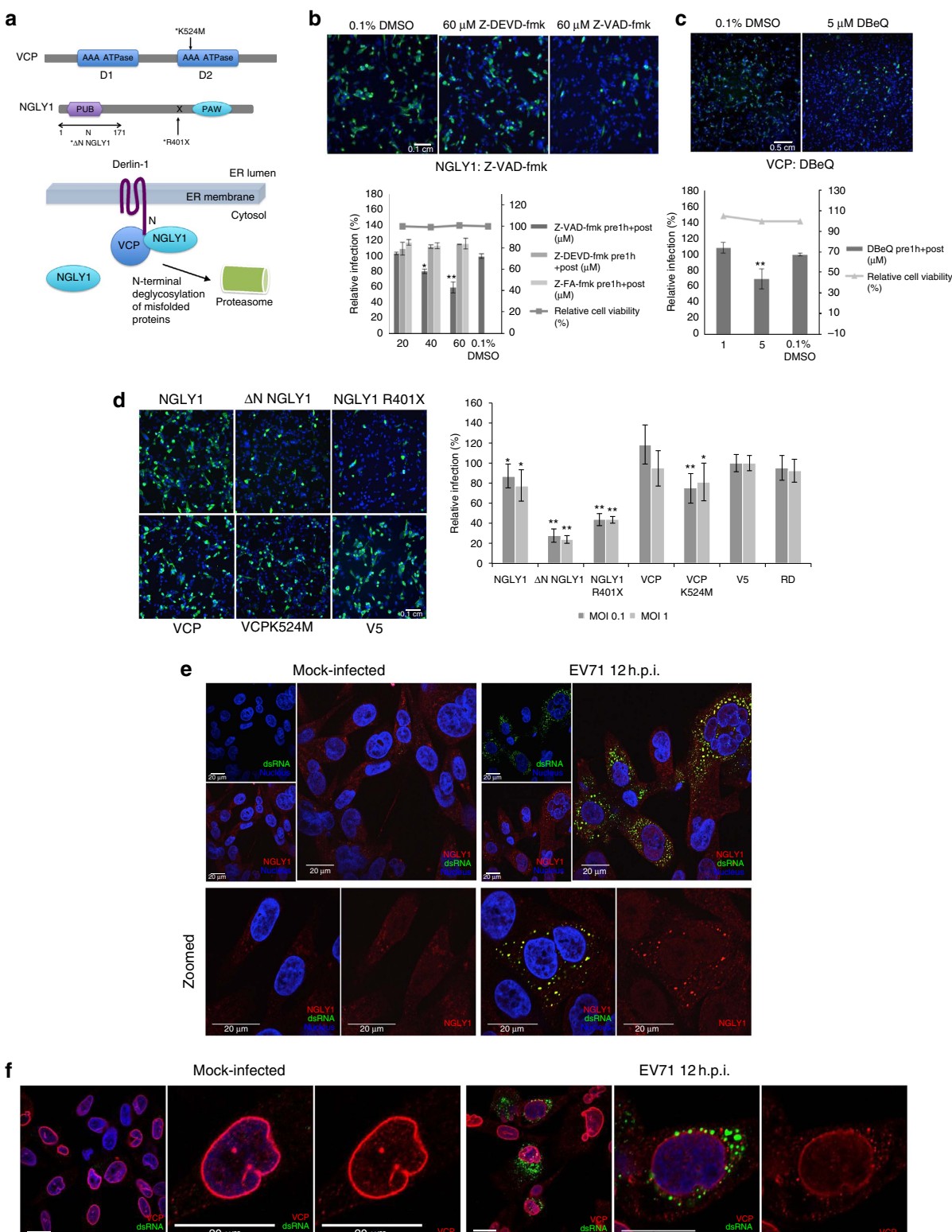

**Figure 4 | NGLY1 and VCP functions are required for EV71 infection.** (**a**) Functional domains of NGLY1 and VCP and their relative interactions at the ER membrane. (**b,c**) Specific inhibition of NGLY1 and VCP inhibits EV71 replication. Bars represent mean relative infection rates from three-independent experiments, normalized against 0.1% DMSO controls. Error bars show standard deviations. (**d**) EV71 infection in cells stably overexpressing NGLY1, ΔN NGLY1, NGLY1 R401X, VCP or VCPK524M. Bars show average relative infection rates measured against V5 (empty vector) expressing cell line and error bars represent standard deviation. Data are from two-independent experiments. (**e**) Immunofluorescence tracking of NGLY1 or (**f**) VCP in mock-infected RD cells and EV71-infected RD cells at 12 h.p.i.. Red signal indicates (**e**) NGLY1 or (**f**) VCP, green for dsRNA and blue for nuclei. Two-tailed, Student's *T*-test was performed against controls, *P < 0.05 and **P < 0.001. Representative scale bars for all micrographs in (**b,d**) can be found in the controls 0.1% DMSO and V5 micrographs respectively, with an indicated length of 0.1 cm. The representative scale bars for (**c**) can be found in the control 0.1% DMSO micrograph with an indicated length of 0.5 cm.

AURKB DN, which resulted in decreased EV71 replication, has a shorter cycling period with cells completing and re-entering S phase after 8 h of exposure to full serum media (Supplementary Fig. 5b). Similarly, the overexpression of CDK6 that resulted in reduced EV71 replication also showed a shortened cycling period. CDK6 overexpressing cells exit and re-enter G2/M phase more rapidly ($\sim$8 h) than V5 or CDK6 DN expressing cells ($\sim$20 h) (Supplementary Fig. 5c). These results suggest a trend where an extended dwell time at specific cell cycle stages aids EV71 replication although further studies will be needed to identify the specific cellular factors or conditions that are involved during these stages.

Although not evaluated individually in downstream experiments, several other genes controlling cell proliferation and cell cycle progression pathways were also identified from our screen as EV71 host factors, such as SKP1, NDC80, MIS12, NEDD9, ZWILCH and FGF8 (Supplementary Data 1). To verify these findings, we used serum-deprivation and small molecule inhibitors to arrest RD cells at specific cell cycle stages and evaluated their capacity in supporting EV71 infection[33]. Serum-deprivation halts cell proliferation and arrests cells at interphase (G0) or early G1. We found a slight but significant decrease ($\sim$20%, $P < 0.05$, Student's $T$-test) of EV71-infected cells in serum-free cultures of RD G1/S phase arrest, whereas hydroxyurea or aphidicolin treatment resulted in a dose-dependent increase in EV71 infection rate ($\sim$20%, $P < 0.05$, Student's $T$-test). Similarly, G2/M phase arrest with nocodazole resulted in significant increase in EV71 infection ($\sim$40%, $P < 0.001$, Student's $T$-test) (Fig. 3g). Flow cytometric analyses of propidium iodide-stained cells treated with these compounds confirmed the expected cell cycle arrests for nocodazole, aphidicolin and hydroxyurea (Supplementary Fig. 5a). In addition, the specific inhibitors of AURKB (AZD1152-HQPA) and CDK6 (PD00332291) also resulted in cell cycle arrests at G2/M and G0/G1 respectively, coinciding with the cell cycle stages AURKB and CDK6 are expected to be involved in (Supplementary Fig. 5a). Taken together, these results show that EV71 replication is restricted in non-proliferating cells but is enhanced at specific cell cycle stages of dividing cells (G1/S for CDK6 and G2/M for AURKB, Fig. 3f).

**ERAD components are essential for EV71 replication**. Picornaviruses can complete their entire replication cycle in the cytoplasm by modifying host intracellular membranes into single or double membrane vesicles that serve as viral RNA replication sites. These replication vesicles have been found to contain both ER and Golgi markers and accumulate as infection progresses to form multilamellar structures[34]. In addition, EV71 can also induce ER stress during infection but modulate its outcome by preventing activation of ERAD through the IRE1-XBP1 pathway[35]. ERAD is a cellular quality control process that removes misfolded proteins from the ER into the cytoplasm for proteasomal degradation[36]. It also serves as an important tool for regulating ER resident enzymes such as HMG CoA reductase, HMGCR[37]. From our screen, proteins participating in ERAD and proteasome functions were identified to be HSFs in EV71 infection.

N-glycanase (NGLY1) is a cytoplasmic protein that is recruited to the ER surface, through its interactions with ER protein Derlin-1 (DER1), to remove N-linked glycans from translocated proteins prior to proteasome loading and degradation[38]. siRNA knockdown of NGLY1 restricted the infection of EV71 in both RD and SK-N-SH cells and also inhibited the replication of CA16 in RD cells (Fig. 2). We verified these findings using the pan-caspase inhibitor, Z-VAD-fmk, that has also been reported to inhibit NGLY1 irreversibly[39]. Significant dose-dependent

inhibition of EV71 infection was found for Z-VAD-fmk but not for other apoptosis inhibitors, Z-DEVD-fmk and Z-FA-fmk, thus ruling out non-specific effects through apoptosis inhibition (Fig. 4). The deletion of the N-terminal region containing the PUB (Peptide: N-glycanase/UBA or UBX-containing proteins) domain of NGLY1 has been shown to prevent NGLY1's interaction with DER1 and hence recruitment to the ER[38]. The C-terminal PAW domain of NGLY1 has been characterized to be a mannose-binding module that increases NGLY1's affinity with its target substrate, glycoproteins (Fig. 4a)[40]. The deleterious mutation, p.R401X, results in a truncated NGLY1 lacking the PAW domain and has been identified in patients presenting with severe NGLY1 deficiencies[41]. The overexpression of either ΔN-NGLY1 or R401X NGLY1 both resulted in a significant reduction of EV71-infected cells by more than 50% ($P < 0.001$, Student's $T$-test). In comparison, wild-type NGLY1 resulted in a slight inhibition of 10–20% (Fig. 4d). These results demonstrate that both NGLY1's recruitment to the ER and its glycanase activity are essential to EV71 infection. Indeed, we observed a distinct accumulation of NGLY1 at double-stranded RNA (dsRNA)-enriched vesicles in EV71-infected RD cells, suggesting a functional role for NGLY1 at virus-induced replication vesicles (Fig. 4e). In order to address the potential involvement of proteasomal function downstream, we tested a series of proteasomal inhibitors but found no significant inhibitory effects on EV71 replication in RD cells (Supplementary Fig. 6a–d). Proteasomal activity assays demonstrated successful inhibition of proteasomal functions at the concentrations tested (Supplementary Fig. 6e–g).

The AAA-ATPase VCP or p97 functions in retro-translocating proteins from the ER for proteasomal degradation and has also been characterized to interact with NGLY1 (Fig. 4a)[42]. A previous study has identified VCP as a host factor for poliovirus but not CB3. VCP was described to interact with poliovirus 2BC and 3AB proteins and may function in viral RNA replication at replication vesicles[43]. Similar to Arita et al., we found that VCP's ATPase activity is essential to EV71 replication. Inhibition of VCP with its reversible inhibitor, DBeQ, resulted in significant reduction of EV71-infected cells ($P < 0.001$, Student's $T$-test) (Fig. 4c)[44]. Overexpression of an inactive mutant, K524M, also resulted in slight but significant inhibition of EV71 ($\sim$20%, $P < 0.001$, Student's $T$-test) (Fig. 4d). In contrast, overexpressing wild-type VCP did not affect EV71 replication. In contrast to the strong colocalization observed with NGLY1 and dsRNA, we could only observe an aggregation of VCP punctae proximal to dsRNA-enriched, virus-induced vesicles in EV71-infected cells (Fig. 4f). This observation is consistent with Arita et al. in poliovirus (PV)-infected cells where VCP was observed to colocalize with 2BC or 3AB but not dsRNA and thus proposed to play a temporal role in autophagosome formation in infected cells. Our findings here extend the role of VCP to EV71 infection and augment the involvement of ER-stress response in EV71 infections to include NGLY1. Further studies will be needed to clarify the individual and relative roles of NGLY1 and VCP in EV71 infection.

**Factors involved in vesicular transport**. Although not identified as part of overrepresented pathways in our pathway enrichment analyses, several factors identified in the screen still held our interest by virtue of their impact on EV71 replication upon knockdown and their novelty as host factors for EV71. One of these factors, radixin (RDX), is a cytosolic, actin-interacting protein identified as an HSF from the primary screen. The ERM (ezrin, radixin and moesin) family consists of highly homologous proteins that can crosslink membranes with the cytoskeleton and play diverse roles in regulating cell morphology, signalling,

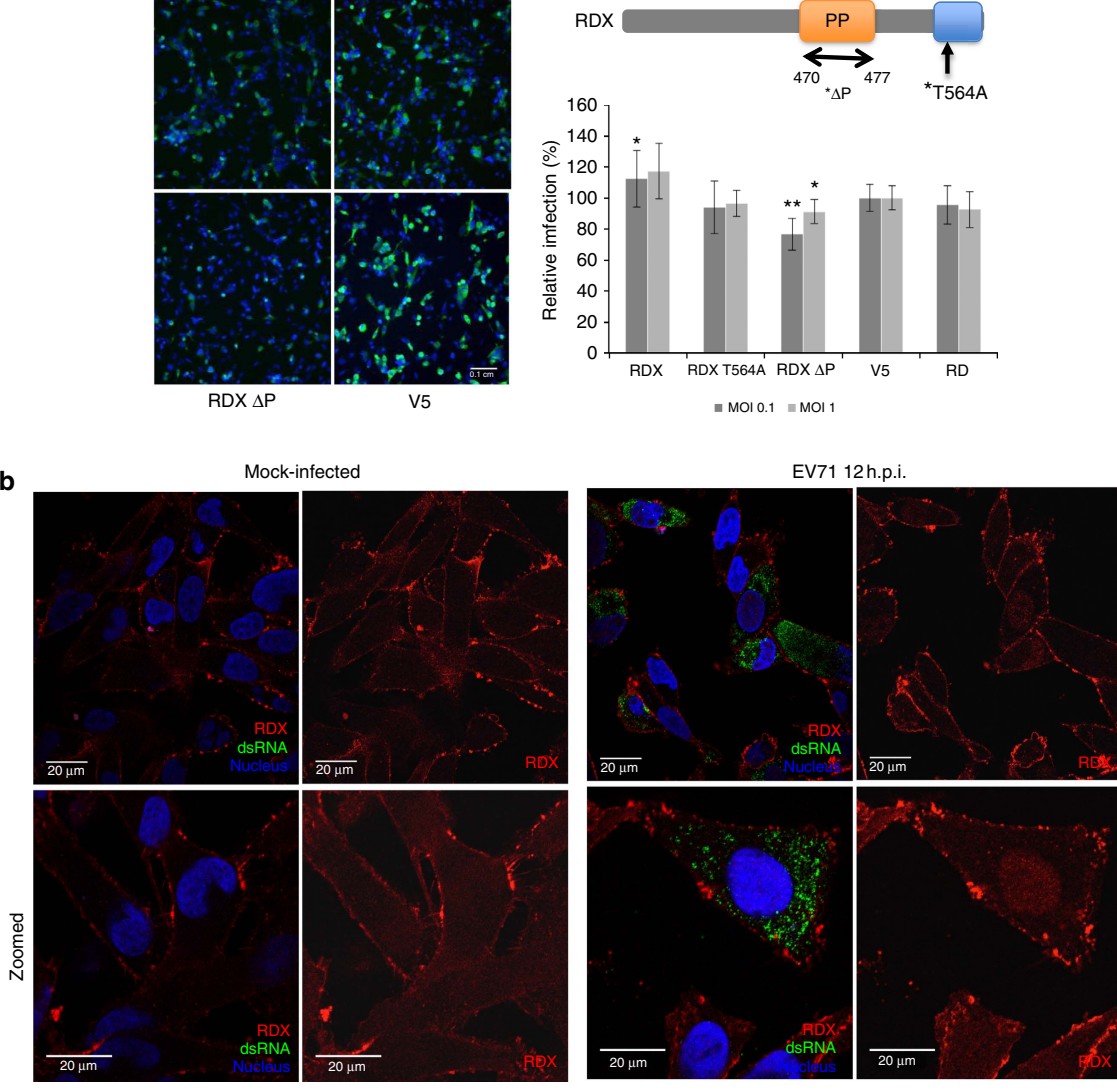

**Figure 5 | RDX is needed for EV71 infection. (a)** Stable cell lines expressing RDX, RDX T564A or RDX ΔP were infected with EV71 and relative infection rates at 12 h.p.i. were compared. Bars show average infection rate normalized against V5 (empty vector) control and error bars represent standard deviation. Data are obtained from two-independent experiments and two-tailed, Student's T-test was performed against controls, *$P < 0.05$ and **$P < 0.001$. Representative scale bar for all micrographs in this panel can be found in the V5 micrograph with an indicated length of 0.1 cm. **(b)** RDX was tracked in immunofluorescence microscopy in mock-infected or EV71-infected RD cells at 12 h.p.i. Red signal indicates RDX, green for dsRNA and blue for nuclei.

membrane trafficking and even cytokinesis[45,46]. Inactive RDX exists in a folded, head-to-tail conformation that obscures its integral membrane protein-binding (N-terminal) and actin-binding (C-terminal) domains (Fig. 5a). Phosphatidylinositol 4,5-bisphosphate (PIP2) binding followed by phosphorylation by kinases at threonine (T564) releases RDX for functional interactions[47].

RDX knockdown significantly inhibited EV71 infection in both RD and SK-N-SH cells. Similar inhibition was also observed in CA16 infection (Fig. 2). Immunofluorescence staining revealed a largely circumferential distribution of RDX in mock-infected RD cells with increased intensity at cell-cell junctions, reflecting RDX's localization to focal-adhesion junctions. While a majority of RDX in EV71-infected cells retained a cell peripheral distribution, a discernable perinuclear fraction of RDX could also be observed (Fig. 5b). The overexpression of wild-type RDX in RD cells resulted in a slight increase (~20%) in EV71 infection rate while the overexpression of mutant RDX lacking a polyproline region (RDX ΔP) resulted in significant reduction in

EV71-infected cells (~20%, $P < 0.001$, Student's T-test) (Fig. 5a). The overexpression of the non-phosphorylatable mutant, RDX T564A (threonine-alanine substitution) did not result in significant changes in EV71 infection rate. Besides T564, RDX contains at least a second phosphorylation site at threonine 573 (refs 48,49). Hence T564A may not be completely dominant negative in the presence of endogenous RDX. The polyproline region, conserved in ezrin and RDX, is largely uncharacterized but has been reported to serve as a potential RDX functional interaction site[50]. Recent reports also described the involvement of ERM proteins in facilitating cell-to-cell spread and virion maturation of parvovirus[51,52]. Our findings revealed RDX as an important EV71 host factor that likely functions to facilitate EV71 entry and/or replication and is also modulated through relocalization during EV71 infection.

## Discussion

Since the re-emergence of neurotropic EV71 in HFMD outbreaks, extensive research has gone into understanding its pathogenesis

in hopes of finding novel therapeutic targets and managing its disease outcome. By screening a human genome-wide siRNA library in a whole-virus infection phenotypic assay, we discovered several previously unknown restrictive or supportive host factors of EV71 infection and replication. This comprehensive approach led us to identify diverse classes of genes as EV71 host factors and revealed a complex interplay between host cell responses and viral manipulation of its host cell to ensure successful replication.

Several genome-wide siRNA screens have been reported for various human viruses to date and overlaps between data sets of the same virus have been notoriously low due to several factors such as cell line, virus strain and methodology[53]. We compared our data set against the only other human picornavirus screen reported, the druggable genome screen for poliovirus and CB3 host factors by Coyne et al.[25] Despite the differences in virus species, cell lines used and scope of screen, we found at least three shared host factors, AKT2, RGS16 and SGK2. While their current roles in enteroviral infections are unknown, future studies should explore if these factors are involved in enteroviral infection specifically or viral infections in general.

In our current study, we also evaluated the effect of selected factors, identified for EV71, on CA16 replication, a closely related enterovirus, and found a tight conservation of host factors. Future work is necessary to explore the extent of conservation of host factors among other HFMD-causing enteroviruses and picornaviruses in general. While we also showed a conservation of host factors across cell lines (Fig. 2), we observed a more pronounced effect on EV71 infection upon knockdown of all genes evaluated in the neuronal cell line, SK-N-SH. Western blot analyses of some of the host factors after siRNA knockdown revealed that knockdown efficiency is gene dependent (Supplementary Fig. 7). The growth kinetics of EV71 is significantly faster in RD cells than SK-N-SH, with peak titre reached by 12 h.p.i. in RD cells while SK-N-SH showed a gradual increase from 24 h.p.i. onwards, before peaking 'gently' at 48 h.p.i. (Supplementary Fig. 7). The data shown for SK-N-SH experiments in this study were all obtained at 24 h.p.i. while that for RD were at 12 h.p.i. Hence, we believe the discrepancy in relative infection rate changes after siRNA knockdown is due to the slower kinetics of EV71 in SK-N-SH cells, giving rise to more pronounced effects when host factor depletion disrupts viral replication.

Despite encoding for two of its own cysteine proteases, 2A and 3C, the host cell proteolytic machinery remains a useful tool for EV71 to regulate host proteins. While NGLY1's deglycanase activity has largely been studied in the context of cellular proteins, we found at least seven predicted sites for N-glycosylation on the EV71 polyprotein using the prediction tool NetNGlyc[54]. Four of these sites occur at the P1 region (structural proteins) of the viral polyprotein while the other three can each be found on 2C, 3C and 3D (RNA-dependent RNA polymerase). To date, there has been no report on glycosylated viral proteins for EV71. It will be interesting to investigate if enteroviral proteins are indeed N-glycosylated during viral replication cycle and if so, the role of NGLY1 in processing these proteins. Importantly, the inhibition of EV71 replication using NGLY1 and VCP inhibitors (Z-VAD-fmk and DBeQ respectively) demonstrated a proof-of-concept that some of the identified host factors can serve as viable host targets for antiviral strategies. Z-VAD-fmk has been previously reported to inhibit the release of EV71 in the context of apoptosis inhibition. However, the study failed to take into account the possibility of direct replication inhibition by measuring viral titre in the culture supernatant only[55]. Moreover, the lack of EV71 inhibition observed for Z-FA-fmk and Z-DEVD-fmk in our experiments ruled out apoptosis-inhibition as the mechanism of action for Z-VAD-fmk.

In addition to the discovery of NGLY1 and VCP as EV71 HSFs, we also identified components of the proteasome itself as important host factors. PSMD1 and PSMD2 encode for the non-ATPase subunits of the substrate-binding 19S regulator in the 26S proteasome complex. The 19S regulator recognizes and binds to the multiubiquitin chain of proteins and may also play a role in dislocating transmembrane ER proteins for degradation[56]. Both PSMD1 and PSMD2 were picked up as HSFs from the primary screen (Supplementary Data 1). However, our experiments with a series of proteasomal inhibitors, MG132, bortezomib, MLN2238 and MLN9708, did not result in significant inhibition of EV71 in RD cells (Supplementary Fig. 6). Proteasome activity assays also confirmed that proteasome functions were inhibited at the concentrations of inhibitors used (Supplementary Fig. 6e–g). Previous studies have implicated the involvement of ubiquitin-proteasome system (UPS) in EV71's regulation of host factors. Some of these studies noted potent inhibition of EV71 by MG132 treatment[57,58]. Interestingly, these findings were demonstrated exclusively in EV71 infection of Vero cells. Similar studies using MG132 in RD cells demonstrated at most, a delay in viral protein synthesis that was not significant by late infection or had no effect on EV71 replication[59,60]. These observations are consistent with our results measuring EV71 infection in RD cells at a 12 h endpoint. The apparent contradictions with MG132 treatment may be due to cell-line-specific differences in proteolytic capacity and should be noted in future studies of UPS in the context of EV71 infection.

Proteolysis through autophagy is an alternative pathway to ER-stress relief and the formation of autophagosome-like vesicles has been reported for EV71 and picornaviruses in general[61–63]. Both NGLY1 and VCP also function in autophagosome formation[64,65]. Our screen identified the core components of the autophagy regulatory complex, BECN1 and PIK3C3 (VPS34), as EV71 HSFs (Supplementary Data 1)[66]. BLOC1S1, a component of the BLOC1 complex functioning in endosomal-lysosomal biogenesis, is also one of the top novel hits identified and verified in primary and secondary screens (Supplementary Data 1)[67]. These results strongly imply the involvement of autophagy and suggest that the roles for NGLY1 and VCP in EV71 infection may also implicate their functions in autophagy. Considering the overlapping functions of autophagy and UPS degradation, future studies will need to delineate their specific contributions towards EV71 replication.

Enhanced infection in replicating cells and the manipulation of cell cycle events has been reported for a number of virus infections, including CB3 (ref. 68). Transcriptomic analyses of EV71-infected RD cells revealed the downregulation of transcripts for CDK6 and CHK1, a spindle checkpoint protein that activates AURKB (ref. 22). A similar study in a neuro-blastoma cell line, SH-SY5Y, also noted an overrepresentation of cell cycle genes that are differentially regulated during EV71 infection[23].

However, it was surprising to find the overexpression of AURKB DN having an inhibitory effect on EV71 replication instead of AURKB overexpression, considering the findings of AURKB as an HRF from siRNA and small molecule studies (Fig. 3). These results could be due to the reported differences in phenotypes of AURKB depletion and expression of dominant-negative AURKB K-R mutant. When AURKB is depleted by siRNA, normal kinetochore assembly can still be observed at the chromosomes during prometaphase. In metaphase, kinetochore becomes disorganized, displaying merotelic attachment that has to be corrected and thus delaying proper chromosome segregation during anaphase and ultimately G2 and M phases' progression[69,70]. In contrast, the expression of the dominant-

negative AURKB (K106R) mutant may result in a hastened G2/M phase, whereby non-functioning AURKB results in the lack of kinetochore attachment to mitotic spindle, causing cells to exit mitosis without displaying anaphase[69]. These previously described, contrasting effects of AURKB knockdown and dominant negative mutation on G2/M phase progression would be consistent with our results, where prolonged or arrest at G2/M phase is perhaps beneficial to EV71 replication.

Together, our results here are consistent with the observations of enhanced virus replication at specific cell cycle stages and went further to demonstrate functionally, the impact some of these factors regulating cell cycle have on EV71 infection. The increased replication observed at G2/M phase coincides with the cellular switch to internal ribosome entry site-dependent translation during mitotic progression[71]. The availability of internal ribosome entry site-transacting factors and other factors during this unique cellular phase may provide an optimal environment for enteroviral replication. Although previous studies with CB3 have proposed a similar hypothesis, the exact mechanism that allows this switch remains unknown[68]. Here we highlighted AURKB and CDK6 as functional host factors for EV71 that can serve as the basis for future work in elucidating the mechanism.

In summary, our genome-wide screening approach has yielded an extensive and diverse collection of host factors and pathways that were previously unreported for EV71. These factors can form the basis for the mechanistic understanding of EV71 pathogenesis in future studies and serve as viable targets for therapeutic development.

## Methods

**Cells and viruses.** RD (CCL-136, ATCC) and SK-N-SH (HTB-11, ATCC) cells were cultured in DMEM supplemented with 10% fetal calf serum (FCS). EV71 (Accession no.: AF316321.2) and CA16 (Accession no.: U05876) used in this study were both propagated in RD cells in reduced serum (2% FCS) DMEM and the stock titres were determined using viral plaque assay with RD cells. Infections were carried out at indicated MOIs (plaque forming unit per cell) for each experiment in 2% FCS DMEM. Experimental endpoints for infection of EV71 or CA16 were fixed at 12 h.p.i. in RD cells and 24 h.p.i. in SK-N-SH to reflect the viral production peaks in each cell type. Cells of similarly low passage counts were used for independent replicate screens to reduce variability. Cells were tested negative for mycoplasma contamination using MycoAlert (Lonza). All live cell cultures performed in this study were incubated in a 5% $CO_2$, 37 °C, humidified incubator.

**Antibodies.** Mouse monoclonal anti-EV71 (MAB979, Millipore, immunofluorescence/IF: 1:1000; Western blot/WB 1:5000) detects VP0/VP2 and cross-reacts with CA16. Rabbit polyclonal anti-CDK6 (PA5-27978, IF 1:200; WB 1:500) and anti-VCP (PA5-22257, IF 1:200) were obtained from Pierce, ThermoScientific. Mouse monoclonal anti-AURKB (ab3609, WB 1:1000), rabbit polyclonal anti-NGLY1 (ab73984, IF 1:200) and anti-RDX (ab52495, IF 1:200; WB 1:500) were obtained from Abcam. Mouse monoclonal anti-dsRNA antibody (J2, IF 1:500) was from Scicons while secondary antibodies, goat fluorescein isothiocynate (FITC)-conjugated anti-mouse antibody (IF 1:500) and goat Rhodamine-conjugated anti-rabbit antibody (IF 1:500) were from Millipore.

**Small molecule inhibitors.** Hydroxyurea (H8627), aphidicolin (A0781), nocodazole (M1404), AZD1152-HQPA (SML0268), PD0332991 isethionate (PZ0199) and DBeQ (SML0031) were all obtained from Sigma-Aldrich. Z-VAD-fmk (03FK10901) was obtained from MPbio, Z-FA-fmk (ab141482) was from Abcam and Z-DEVD-fmk (2166) was from Tocris. Proteasome inhibitors, bortezomib (S1013), MG132 (S2619), MLN2238 (S2180) and MLN9708 (S2181) were all sourced from Selleckchem. With the exception of PD0332991, which was dissolved in water, stock solutions for all of the compounds above were prepared in dimethyl sulfoxide (DMSO), aliquoted and stored at − 80 °C.

**High-throughput RNAi screening.** The siRNA library consisting of pools of four siRNA sequences per gene (siGENOME, ThermoScientific) was spotted onto 384-well, clear bottom black plates (Bio-one, Grenier) while the control siRNA pools were dispensed into each plate at the indicated well positions (Fig. 1a). The addition of solutions was all carried out using a liquid handler, Multidrop Combi (ThermoScientific), to minimize variability. The transfection reagent, Dharmafect-1 (0.125 µl in 20 µl DharmaFECT cell culture reagent (DCCR),

ThermoScientific) was dispensed into each well and left to incubate at room temperature for 30 min to allow for complexing to take place. Using reverse transfection, RD cells (5000 cells in 30 µl) were then added into each well to give a final siRNA concentration of 25 nM. The transfected cells were incubated for 72 h for siRNA knockdown before virus infection. Briefly, culture media in each well was aspirated using an automated plate washer (AquaMax 2000, Molecular Devices) and 10 µl of virus suspension (MOI 1) was dispensed into each well and incubated for 1 h for adsorption to take place before the wells were topped up with fresh 2% FCS DMEM. After a further incubation for 12 h post infection, the cells were fixed in a final concentration of 4% paraformaldehyde-phosphate buffered saline (PBS) solution with 0.1% Triton-X (detergent for permeabilization) for 10 min at room temperature. The fixed cells were then washed with PBS, incubated with anti-VP0/VP2 antibody (1:1000) followed by anti-mouse FITC-conjugated secondary antibody and finally 4′,6-diamidino-2-phenylindole (DAPI). The secondary screen follows an identical protocol with the exception of the ON-TARGETplus series of siRNA pools replacing siGENOME pools for target genes and controls. Deconvolution and validation screens for CA16 infection in RD cells and EV71 in SK-N-SH cells were carried out in 96-well black optical plates (BD Falcon) using single siRNA per gene. Assay conditions for 96-well were scaled-up accordingly for 20,000 cells reversed transfected with 25 nM siRNA and 0.5 µl of Dharmafect-1 per well.

**Automated imaging and data analyses.** Images were captured at four sites per well at × 10 for 384-well plates and nine sites per well at × 20 for 96-well plates using ImageXpress Micro (Molecular Devices). Infection rate was obtained using object identification modules in CellProfiler to enumerate the number of DAPI-stained nuclei and FITC-antigen expressing cells[72]. Data processing and screening quality checks that include data spread and correlation between screens were performed using Screensifter[73]. For primary screen results, raw data for each gene were normalized against the mean and standard deviation of infection rates from all sample genes in its residing plate using the formula $Z = (X - \text{Ave}(X_{pl}))/(\text{SD}(X_{pl}))$. The average Z-score from independent replicate screens was then determined for each gene prior to hit identification. The hit identifying thresholds were set with reference to SCARB2 (HSFs, $Z_{av} < Z_{SCARB2} + 2\text{SD}_{SCARB2}$) and NT controls (HRFs, $Z_{av} > Z_{NT} + 2\text{SD}_{NT}$). The hits from the primary screen are listed with the average normalized Z values and the standard deviations from three-independent screens (Supplementary Data 1). The hits removed due to the cytotoxic cutoff are also shown (Cytotoxic hits column).

To account for the bias in the curated secondary screen, normalization was performed using SCARB2 and NT controls of each plate with the formula, $Ctrl_{NORM} = (X - \text{Ave}(X_{SCARB2}))/(\text{Ave}(X_{NT}) - \text{Ave}(X_{SCARB2}))$. Hit determination was set at $Ctrl_{NORM} < 0.60$ for HSFs and $Ctrl_{NORM} > 0.92$ for HRFs based on observations of data distribution (Supplementary Fig. 3). Hits were classified using Panther classification system[74] and pathway analyses were carried out using the Reactome plugin for Cytoscape[75,76].

**Immunofluorescence staining and confocal microscopy.** Cells were seeded onto glass coverslips and treated as described. At indicated timepoints, samples were fixed with 4% paraformaldehyde at room temperature for 10 min before permeabilization with 0.1% Triton-X at room temperature for 5 min. The fixed samples were rinsed blocked with 1% bovine serum albumin solution at 37 °C for 1 h and incubated with primary antibodies overnight at 4 °C before three consecutive washes in PBS to remove excess antibodies. Incubation with secondary conjugated antibodies was carried out for 1 h at 37 °C before coverslips are washed and mounted onto a glass slide with anti-fade mounting medium containing DAPI (Sigma-Aldrich). Slides were imaged using Nikon A1R⁺ si confocal microscope at Nikon Imaging Center (Singapore Bioimaging Consortium).

**Generation of stable cell lines.** The open reading frames (ORFs) for each gene were cloned into a common vector, pLX304, which contains a C-terminal V5 tag and a blasticidin-resistance selection marker. The original ORFs for each gene were sourced from either the CCSB-Broad Lentiviral Expression Library or the NIH-Mammalian Gene Collection (MGC). Access to these libraries was kindly provided by Asst. Prof. Krishnan Manoj (Duke-National University of Singapore (NUS)) and Prof. Naoki Yamamoto (NUS). ORFs sourced from the NIH-MGC collection are CDK6 (BC052264), AURKB (BC000442), DTX1 (BC048216), VCP (BC110913), PMVK (BC007694) and TGFβ1I1 (BC017288). The remaining genes were cloned from the CCSB-Broad collection; RDX (ccsbBroad304_06855), NGLY1 (ccsbBroad304_03651), CNOT3 (ccsbBroad304_06648), SNUPN (ccsbBroad304_02304), LMO1 (ccsbBroad304_06530). Site-directed mutagenesis was performed using In-Fusion HD cloning (Clonetech) and verified by sequencing. Transfection was performed using jetPrime transfection reagent (Polyplus-transfection) with 2 µg of plasmid to 200,000 RD cells in a 6-well dish. Selection for stable expressing population was performed using 10% FCS DMEM supplemented with 5 µg ml⁻¹ blasticidin (Life Technologies).

**Western blot.** Western blot samples were prepared by lysing PBS-washed cells in 1 × Laemmli sample buffer at 4 °C for 10 min with gentle rocking before harvesting the lysate and heating it at 95 °C for 10 min. The protein samples were

resolved in 10% SDS-polyacrylamide gels before transferring to nitrocellulose membranes and blocked with 5% skimmed milk in tris-buffered saline with Tween 20 solution. The blots were then probed with primary antibodies overnight at 4 °C with constant agitation before washing and detection with secondary antibodies, goat horseradish peroxidase-conjugated anti-mouse (1:10,000) or anti-rabbit (1:10,000) from Millipore. Signal development was achieved using enhanced chemiluminescent substrate (Supersignal West Dura, Thermo Scientific). The blots were scanned with C-DiGit (Licor) and band intensities were determined with Image Studio (Licor). Uncropped scans of the Western blots for AURKB, CDK6 and RDX can be found in Supplementary Fig. 7c.

**Quantitative RT-PCR (qRT-PCR).** Infected cells were lysed at the indicated timepoints and total RNA was extracted using a Total RNeasy kit (Qiagen). EV71 RNA was detected using a SYBR green-based RT-PCR kit (Maxima, Thermo Scientific) in the StepOne Plus Real-time PCR system (Applied Biosystems). The primers used in targeting the 5′ UTR of EV71 were MD90 (5′-ATTGTCACCA TAAGCAGCCA-3′) and MD91 (5′-CCTCCGGCCCCTGAATGCGGCTAAT-3′). Actin was detected simultaneously as a housekeeping gene representing total RNA material sampled and the sequences for the forward and reverse primers were 5′-AGCGCGGCTACAGCTTCA-3′ and 5′-GGCGACGTAGCACAGCTTCT-3′ respectively.

**Proteasome activity assay.** Cells were treated with inhibitors as described in figures and harvested for proteasome activity analysis according to instructions provided in the fluorometric proteasome activity assay kit (J4120, UBPBio). Readings were obtained using a Tecan Infinite 200 reader.

**Propidium iodide staining and flow cytometry.** Cells were treated as described in figures and trypsinized for fixation in ice-cold, 70% ethanol overnight before staining with FxCycle PI/RNase Staining Solution (Molecular Probes) according to the instructions provided and analysed using CyAn Advanced Digital Processing High-Performance Flow Cytometer (Beckman Coulter).

**Data availability.** All relevant data are available from the authors on request.

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

## Acknowledgements

We would like to thank Asst. Prof. Krishnan Manoj (Duke-NUS) and Prof. Naoki Yamamoto (NUS) for providing us with the ORF clones from their library collection. This work was supported by grants from MINDEF DIRP 2011 (POD0001476) and the National Medical Research Council (IRG11may096 and CBRG13nov023).

## Author contributions

J.J.-H.C. and F.B. conceptualized the study. K.X.W. and P.P. designed the research and experiments. K.X.W. and P.P. conducted the experiments and data analyses. D.M., P.K., G.Y.L.G. and F.B. advised and assisted in screening design and setup. P.K. provided ScreenSifter software and assisted with the analysis of screening data. V.T.K.C. and F.B. contributed necessary reagents and protocols. J.J.-H.C. and F.B. supervised the study. K.X.W. and J.J.-H.C. wrote the manuscript. Funding for study was acquired by J.J.-H.C.

## Additional information

**Competing financial interests:** The authors declare no competing financial interests.

**How to cite this article**: Wu, K. X. *et al.* Human genome-wide RNAi screen reveals host factors required for enterovirus 71 replication. *Nat. Commun.* **7,** 13150 doi: 10.1038/ncomms13150 (2016).

