## [Peer Review File · Nature Communications]

Reviewers' comments:

Reviewer #1 (Remarks to the Author):

In this manuscript, the authors identified host factors that affect EV71 infection by using RNAi screening. They have knocked down the genes in RD cells and picked up 517 genes. They then validated these hits using different siRNAs. They also evaluated the effect of the 16 genes for EV71 infection in SK-N-SH cells and for CVA16 infection in RD cells. The authors characterized some of the genes.

The aim of report is important for understanding of EV71 replication cycle and searching of anti viral targets. This is the first report of genome-wide siRNA screening.

However, this reviewer feels several points are not enough to support author's claims. The presentation of the results and materials used are not kindly explained. Specific comments are written below.

1. The title of the manuscript is not adequate. The authors have identified the host factors required for EV71 REPLICARION but not pathogenesis. In addition, the authors only tested the effect of knockdown of these genes on EV71 and CVA16. It is not clearly written whether they had also effect on other picornaviruses or unrelated viruses. The authors should show it.
2. The authors should show the knockdown efficiency by western blotting.
3. They have selected several candidate genes for HSF and HRF. The related infection efficiency (Fig 3A) and virus titer (supplementary Fig. 4) in cells knocked down with these genes looked not significantly different from those with NT, while knock down of SCARB2 had a very clear effect. The authors wrote "the knockdown of the CDK6 and AURKB resulted in increased EV71 replication in both RD and SK-N-SK cells (see Fig3)" (lines 197-198). This is not true. Significant increase is observed only in AURKO knockdown SK-N-SK cells. The authors should show the results of knock down experiments of host factors that had been reported to be involved in EV71 replication by other methods.
4. These host factors identified by the authors include cell cycle and transcription regulators. This reviewer feels that knockdown of cell cycle or transcription regulators causes global effects for the cells. The authors should be careful if these factors have direct effect or indirect effect. If the effect is indirect, it is really adequate to conclude that the factors are HSF or HRF?
5. In Fig 4A, the authors seems to use several DTX1 mutants. However, these mutants are not explained in Materials and Methods section and in the legends. Therefore this reviewer had a difficulty in understanding lines 170-186.
6. In Fig 5B, there is a photo of V5 without any explanation. What does it mean?
7. Overexpression experiment of AURKB and its dominant negative mutant seemed inconsistent with knockdown experiment. The authors comment on this discrepancy is not sufficient (line 220-231).
8. Overexpression experiment of PMVK also seemed inconsistent with knockdown experiment. Experimental evidence is needed to explain this discrepancy.
9. In lines 322-337, the authors suggested that CD1a is involved in entry step. However, no experimental data is presented. The author should either show the experimental evidences or delete these sentences.

10. Some of the sentences in Discussion section are not related to the experiments shown in this manuscript. I suggest to discuss specific points in the experiments in this manuscript.

Reviewer #2 (Remarks to the Author):

The manuscript by Kan Xing Wu and co-workers reports the results of a genome-wide RNAi screening aimed at identifying host cellular factors controlling infection by Enterovirus 71. By performing a fluorescence microscopy based screening, the authors identified a total of 905 candidate host genes with a strong effect on infection, specifically 715 host susceptibility factors and 190 host resistance factors. Gene ontology and pathway analysis revealed the involvement of both known and unexpected processes in the interaction of Enterovirus 71 with host cells. Partial validation of this candidate gene list through a secondary screening (517 genes subjected to validation) yielded a final list of 257 validated hits. Subsequently, the authors focused on a small subset of genes and performed a series of additional experiments meant to demonstrate the relevance of the identified targets in several cellular processes. This include the study of DTX1, CDK6 and AURBK in cell proliferation and cell cycle, NGLY1 in ER-associated degradation AD, as well as the involvement of the proteasome, mevalonate pathway and vesicular transport in Enterovirus 71 infection.

Overall, the screening experiments are well conducted and constitute a valuable resource for researchers working in the field of host-pathogen interactions. Notwithstanding, the manuscript lacks focus and, by trying to explore the involvement of different genes in distinct cellular processes, the authors fail to produce convincing evidence for most of these.

Data interpretation is in general superficial and contradicting evidence obtained in overexpression or fluorescence microscopy experiments is not addressed properly, but rather taken as a confirmation of the involvement of the genes in question in the infection by Enterovirus 71. Important controls (e.g. effect of CDK6 inhibitors and constructs on cell cycle, functional inhibition of proteasome) are missing and should be included.

The workflow of the screening and overall results are spread along Figures 1, 2 and 3. This data could easily be shown in one main figure since a significant part of the information provided therein should be considered supplementary material (e.g. Figs 1B, 1C, 2; figure 1D should be removed). In general, the data are not presented clearly, particularly the validation experiments; statistics are missing in several panels (e.g. 3a, 3b); several panels are not referred to in the text (e.g. S1b, S1c).

Given the low validation rate of the candidate genes selected from the primary screening, it is questionable that gene ontology and pathway analysis was performed on this gene list, which arguably contains 50% false positives.

In the opinion of this Reviewer, the data obtained from the screening is of excellent quality, but the authors should restrict their analysis to the involvement of the newly identified hits on one or two cellular processes, providing in depth analysis of the phenotypes. The detailed follow-up of other interesting hits should be the subject of other future works.

Reviewer #3 (Remarks to the Author):

The authors performed a siRNA-based genome wide screen (an immunofluorescence-based phenotypic screen against a genome-wide siRNA library of ~21.000 human genes) to identify or confirm host susceptibility factors and host resistance factors, which respectively facilitate or suppress EV-A71 infection in RD cells. They identified and validate 256 host factors and they carried out functional validation of 16 of the 256 hits in the human neuronal cell line SK-N-SH (also in the context of infection with another enterovirus i.e. the Coxsackie virus A16). These host factors were found to be involved in either cell cycle control, the ERAD pathway, mevalonate and de novo lipid synthesis and vesicular transport. A small molecule inhibitor of one of the factors essential for viral replication, i.e. NGLY1, also reduced EV-A71 replication. Overall, this interesting and well performed study exhaustively provides the first map of EV-A71-host interaction and reveals potential antiviral targets.

Major comments

This paper represents a lot of work and contains also an almost overload of information. This makes it rather difficult to read and digest. I advise to delete the preliminary findings reported in the manuscript (even though interesting) and also to largely reduce the length of the discussion. The study would also benefit from a comparison with other studies (for example the work by Coyne and colleagues).

Line 87: It is very important to show the kinetics of in vitro infection of the EV-A71 clinical isolate used. An end-point titration up to 48 hr post infection (also for the CV-A16 infection) will be needed to know the status of the infection at the time of the screening (12h) and this either in presence or absence of the siRNA-control pool. The fact that host factors involved in cell cycle control and the PI3K survival pathway are being identified but not factors involved in for example apoptosis and autophagy induction may be related to the selection of the time-point (besides the comments already made in the discussion (line 407))

Line 152: the authors do not comment on the fact that the relative infection ratio between control cells and silenced cell (no matter the gene) is greater in SK cells than in RD cells. Does it depend on the greater efficiency of transfection and silencing in the neuronal cell line? It will be important to demonstrate the silencing efficiency in both cell lines has by for example Western blotting or qRT-PCR and this at least for the 16 hits that were selected for further study.

Fig.4a The figure is confusing. Some factors are not mentioned in the text and do not show complementary behavior compared to figure 3a (for instance, the overexpression of PMVK and CNOT3 do not increase the relative infection of EV-A71 even though their silencing decreases EV-A71 relative infection). By consequence, the finding on DTX1 lacking the RING finger is not informative in absence of the full-length DTX1 control. Overall the findings on DTX1 are too preliminary to be presented and given the density of results reported in the

paper, this part can be eliminated without affecting the integrity and impact of the paper (see also comments above). Same remark for the paragraph 'disrupting the mevalonate pathway inhibits EV71 replication', even though the authors try to explain the counterintuitive effect of the overexpression of PMVK, no additional results are presented in this paragraph. Hence this section may be omitted though some brief mentioning of these findings in the discussion may be relevant.

Minor comments

-Line 71: reference Coyne et al missing

-Line 274: the authors take for granted that NGLY1 recruitment and glycanase activity are dependent on the PUB domain and the R401X mutation. This has to be stated in the text.

-Line 407: 'host transcription is necessary for enterovirus replication at early time point of infection' is an old finding ('70s). The best demonstration is the effect of actinomycin D in the context of poliovirus infection, which decrease viral yield at early times post-infection.

Reviewers' comments:

Reviewer #1 (Remarks to the Author):

In this manuscript, the authors identified host factors that affect EV71 infection by using RNAi screening. They have knocked down the genes in RD cells and picked up 517 genes. They then validated these hits using different siRNAs. They also evaluated the effect of the 16 genes for EV71 infection in SK-N-SH cells and for CVA16 infection in RD cells. The authors characterized some of the genes.

The aim of report is important for understanding of EV71 replication cycle and searching of anti viral targets. This is the first report of genome-wide siRNA screening.

However, this reviewer feels several points are not enough to support author's claims. The presentation of the results and materials used are not kindly explained. Specific comments are written below.

We thank the reviewer for the careful reading and constructive criticisms given. We have taken these points into serious consideration and revised the manuscript accordingly. Specific comments are also addressed below.

1. The title of the manuscript is not adequate. The authors have identified the host factors required for EV71 REPLICARION but not pathogenesis. In addition, the authors only tested the effect of knockdown of these genes on EV71 and CVA16. It is not clearly written whether they had also effect on other picornaviruses or unrelated viruses. The authors should show it.

We thank the reviewer for the suggestion and have changed the title to “Human genome-wide RNAi screen reveals host factors required for Enterovirus 71 replication”.

We have yet to test any of the genes against picornaviruses beyond EV71 or CA16 as that is not within the scope of the current study. Future work will definitely look into other HFMD-causing picornaviruses and unrelated viruses.

2. The authors should show the knockdown efficiency by western blotting.

We have performed Western blots for AURKB, CDK6 and RDX against NT siRNA in both RD cells and SK-N-SH. We have also performed densitometric quantitation to reflect the percentage of knockdown in each cell line.

This was also done to address the concern of Reviewer #3 on the possibility of differential knockdown efficiencies in RD versus SK-N-SH cells that may explain the greater impact of gene knockdowns on EV71 infection rates in SK-N-SH cells. As can be seen from the knockdown of these 3 genes, siRNA knockdown efficiency is variable between genes in either cell lines (Fig S7). As such the consistently greater impact of siRNA knockdown on EV71 infection rates in SK-N-SH cells cannot be explained by knockdown efficiency but is more likely reflective of the slower rate of EV71 replication seen in SK-N-SH cells (Fig. S7). This has been included in our discussion at Lines 641-655.

3. They have selected several candidate genes for HSF and HRF. The related infection efficiency (Fig 3A) and virus titer (supplementary Fig. 4) in cells knocked down with these genes looked not significantly different from those with NT, while know down of SCARB2

had a very clear effect. The authors wrote "the knockdown of the CDK6 and AURKB resulted in increased EV71 replication in both RD and SK-N-SK cells (see Fig3)" (lines 197-198). This is not true. Significant increase is observed only in AURKO knockdown SK-N-SK cells. The authors should show the results of knock down experiments of host factors that had been reported to be involved in EV71 replication by other methods.

We have performed an additional independent repeat of the experiments and also carried out Student's T-test against the NT controls. p-values for each gene knockdown are shown alongside in a table beside each chart for both IF-based infection rates (Figs. 2a and 2c) and plaque assay reductions (Figs. S4b-d). To better represent the differences seen in plaque reduction from single gene knockdowns, we have also presented each average titer obtained as a percentage of that obtained in NT-knockdown samples (Figs. S4b-d). These results will show that there are indeed significant differences in EV71 replication after knockdown of AURKB in both RD and SK-N-SH cells.

We have also carried out qRT-PCR experiments to supplement the IF and plaque assay results for AURKB, CDK6, RDX and NGLY1 knockdowns. Viral RNA levels were significantly increased for the HRFs (CDK6 and AURKB) while knockdown of HSFs resulted in significant decreases in viral RNA when compared to the NT controls (Fig. 2b).

4. These host factors identified by the authors include cell cycle and transcription regulators. This reviewer feels that knockdown of cell cycle or transcription regulators causes global effects for the cells. The authors should be careful if these factors have direct effect or indirect effect. If the effect is indirect, it is really adequate to conclude that the factors are HSF or HRF?

We are aware of the possibility of indirect effects at play with cell cycle factors. However, our own results here along with others published in the literature suggests that cell cycle manipulation or replication bias at specific stages is a recurring phenomenon even for cytoplasm-replicating RNA viruses (see Discussion). Our study picked up at least two specific factors, AURKB and CDK6, which were also found to be modulated at the transcript level in an EV71 transcriptomic study.

The definitions of HSF/HRF here are based entirely on the phenotypic cutoffs set in the study, i.e. relative infection rate based on IF quantitation. We also think that host factors for virus replication extends beyond direct interaction partners or even protein complex components but include factors in signaling pathways and other cellular processes that together are modulated upon virus infection to create a conducive replication environment. However, we do agree that it will be important to delineate the specific factors involved in the identified cell cycle stages in future studies.

5. In Fig 4A, the authors seems to use several DTX1 mutants. However, these mutants are not explained in Materials and Methods section and in the legends. Therefore this reviewer had a difficulty in understanding lines 170-186.

As pointed out by other reviewers, the results for DTX1, CD1a and PMVK are too preliminary and limited at the current stage for inclusion into the manuscript. As such we have removed the data and discussion pertaining to DTX1 and PMVK.

6. In Fig 5B, there is a photo of V5 without any explanation. What does it mean?

We apologize for the lack of explanations but V5 is the empty vector used as controls for all overexpressing studies. We used a V5-tag carrying plasmid vector for all overexpression clones and the data showing V5 represents that arising from cell lines stably-expressing the V5-tag only.

7. Overexpression experiment of AURKB and its dominant negative mutant seemed inconsistent with knockdown experiment. The authors comment on this discrepancy is not sufficient (line 220-231).

We have carried out PI staining and flow cytometry analysis of all overexpressing clones for the cell cycle studies and found a shortened cell cycling period for cell lines that showed reduced EV71 replication, AURKB DN and CDK6. This data complements the observation that extended dwell time in G2/M phase or G1 phase from AURKB or CDK6 inhibition supports greater EV71 replication and the shortening of these cycling phases will inhibit EV71 replication.

8. Overexpression experiment of PMVK also seemed inconsistent with knockdown experiment. Experimental evidence is needed to explain this discrepancy.

As mentioned earlier in response to point 5, we have removed all PMVK relevant content at this stage. Further studies will be done to hopefully clarify this discrepancy.

9. In lines 322-337, the authors suggested that CD1a is involved in entry step. However, no experimental data is presented. The author should either show the experimental evidences or delete these sentences.

As suggested by the reviewers, we have removed parts that are too preliminary and lacking in experimental data from the manuscript. The parts regarding CD1a have been removed from both the Results and Discussion sections.

10. Some of the sentences in Discussion section are not related to the experiments shown in this manuscript. I suggest to discuss specific points in the experiments in this manuscript.

We thank the reviewer for the helpful suggestion and have since edited the Discussion section to hopefully, a more pertinent and succinct version.

Reviewer #2 (Remarks to the Author):

The manuscript by Kan Xing Wu and co-workers reports the results of a genome-wide RNAi screening aimed at identifying host cellular factors controlling infection by Enterovirus 71. By performing a fluorescence microscopy based screening, the authors identified a total of 905 candidate host genes with a strong effect on infection, specifically 715 host susceptibility factors and 190 host resistance factors. Gene ontology and pathway analysis revealed the involvement of both known and unexpected processes in the interaction of Enterovirus 71 with host cells. Partial validation of this candidate gene list through a secondary screening (517 genes subjected to validation) yielded a final list of 257 validated hits. Subsequently, the authors focused on a small subset of genes and performed a series of additional experiments meant to demonstrate the relevance of the identified targets in several cellular processes. This include the study of DTX1, CDK6 and AURKB in cell proliferation and cell cycle, NGLY1 in ER-associated degradation AD, as well as the involvement of the proteasome, mevalonate pathway and vesicular transport in Enterovirus 71 infection. Overall, the screening experiments are well conducted and constitute a valuable resource for researchers working in the field of host-pathogen interactions. Notwithstanding, the manuscript lacks focus and, by trying to explore the involvement of different genes in distinct cellular processes, the authors fail to produce convincing evidence for most of these.

We thank the reviewer for the constructive comments. We have taken on board all the suggestions from the reviewers and edited the manuscript to remove most of the preliminary data and focused mainly on the CDK6/AURKB and NGLY1/VCP studies. We have also performed additional studies to bolster some of the results in these studies. We hope the reviewer will find these changes useful in providing focus to the manuscript.

Data interpretation is in general superficial and contradicting evidence obtained in overexpression or fluorescence microscopy experiments is not addressed properly, but rather taken as a confirmation of the involvement of the genes in question in the infection by Enterovirus 71.

Important controls (e.g. effect of CDK6 inhibitors and constructs on cell cycle, functional inhibition of proteasome) are missing and should be included.

We have modified the manuscript and removed most of the preliminary data involving factors such as CD1a, PMVK, DTX1 and TGFB111. Instead we have focused on factors where we have multiple angles of validation from small molecule inhibitors, overexpression studies and microscopy.

In addition, we also performed flow cytometric cell cycle analysis as suggested by the reviewer and revealed the effects overexpression of CDK6, AURKB or their functional mutants have on the cell cycle. Cell cycle inhibitors and inhibitors of CDK6 and AURKB were also evaluated to confirm their expected arrests at specific cell cycle stages (Fig. S5).

Using a three-substrate proteasome activity assay to test for chymotrypsin-like, trypsin-like and caspase-like activities, we showed that proteasomal activities were inhibited for all proteasome inhibitors at the concentrations used (Fig. S6).

The workflow of the screening and overall results are spread along Figures 1, 2 and 3. This data could easily be shown in one main figure since a significant part of the information provided therein should be considered supplementary material (e.g. Figs 1B, 1C, 2; figure 1D

should be removed). In general, the data are not presented clearly, particularly the validation experiments; statistics are missing in several panels (e.g. 3a, 3b); several panels are not referred to in the text (e.g. S1b, S1c).

We thank the reviewer for the suggestion and have since changed the figure layouts accordingly. Figures 1 and 2 are now combined into a single figure with most of the less important subfigures (1B, C and 2A and B) moved into supplementary. Figure 1D has also been removed completely.

Statistics have now been included for all validation experiments with p-values for each gene displayed in a table beside each bar chart. We hope this presentation style can provide more information while giving a cleaner look to the data by keeping multiple asterisks off the bar charts.

The relevant in text references have been added for these supplementary figures S1d, e, f and g.

Given the low validation rate of the candidate genes selected from the primary screening, it is questionable that gene ontology and pathway analysis was performed on this gene list, which arguably contains 50% false positives.

We understand the concern of the reviewer and the main reason for our use of primary screening hits is due to our selection criteria of hits for secondary screening validation. In the attempt to maximize experimental costs and validate novel and better characterized candidate genes, we dropped genes from the primary screen that have been reported in the literature for EV71 and genes that are too poorly characterized at this stage to be studied meaningfully. As such, this 'censorship' upstream will automatically skew any validated hit list during pathway enrichment analysis and remove reported pathways such as COPI transport and heparan sulfate pathways. By choosing to analyse the primary hit list, we hope to present a more representative and holistic genomic landscape involved in EV71 replication.

It is our view that any gene ontology and pathway analysis are at best advisory and will have to be backed by functional validation studies. However, we understand the concern of misrepresentation and have since removed Fig. S2 and moved Fig. 2A and B into supplementary. We have also removed in-text discussion of any results arising from the analysis of non-validated hits and have retained only discussions of known pathways identified from the analysis that serve to bolster confidence in the screening results.

In the opinion of this Reviewer, the data obtained from the screening is of excellent quality, but the authors should restrict their analysis to the involvement of the newly identified hits on one or two cellular processes, providing in depth analysis of the phenotypes. The detailed follow-up of other interesting hits should be the subject of other future works.

Reviewer #3 (Remarks to the Author):

The authors performed a siRNA-based genome wide screen (an immunofluorescence-based phenotypic screen against a genome-wide siRNA library of ~21.000 human genes) to identify or confirm host susceptibility factors and host resistance factors, which respectively facilitate or suppress EV-A71 infection in RD cells. They identified and validate 256 host factors and they carried out functional validation of 16 of the 256 hits in the human neuronal cell line SK-N-SH (also in the context of infection with another enterovirus i.e. the Coxsackie virus A16). These host factors were found to be involved in either cell cycle control, the ERAD pathway, mevalonate and de novo lipid synthesis and vesicular transport. A small molecule inhibitor of one of the factors essential for viral replication, i.e. NGLY1, also reduced EV-A71 replication. Overall, this interesting and well performed study exhaustively provides the first map of EV-A71-host interaction and reveals potential antiviral targets.

We thank the reviewer for the careful reading of our manuscript and the kind comments and constructive criticisms given.

Major comments

This paper represents a lot of work and contains also an almost overload of information. This makes it rather difficult to read and digest. I advise to delete the preliminary findings reported in the manuscript (even though interesting) and also to largely reduce the length of the discussion. The study would also benefit from a comparison with other studies (for example the work by Coyne and colleagues).

We have taken the suggestion of the reviewer and removed most of the preliminary findings involving DTX1, TGFBI1, CD1a and PMVK and chosen to focus on CDK6/AURKB and NGLY1/VCP studies. We hope this will provide the manuscript with more focus and make for an easier read. We have also compared the data set obtained here with that from the only other human enterovirus screen by Coyne and colleagues. However due to differences in screening library (whole genome and druggable genome), cell line and virus species, we could only find 3 overlapping hits, AKT2, RGS16 and SGK2. These findings have been included in the Discussion section Lines 633-641.

Line 87: It is very important to show the kinetics of in vitro infection of the EV-A71 clinical isolate used. An end-point titration up to 48 hr post infection (also for the CV-A16 infection) will be needed to know the status of the infection at the time of the screening (12h) and this either in presence or absence of the siRNA-control pool. The fact that host factors involved in cell cycle control and the PI3K survival pathway are being identified but not factors involved in for example apoptosis and autophagy induction may be related to the selection of the time-point (besides the comments already made in the discussion (line 407)

We have included the growth kinetics for EV71 (HFM41) in both RD an SK-N-SH cells and that of CA16 in RD cells up to 72hpi (Fig. S7b). The choice for 12hpi for EV71 on RD cells was made based on the peak virus titer obtainable while retaining a relatively intact monolayer of cells that make it possible for IF processing and imaging. As we did not perform endpoint viral titer analysis (plaque assay) from each gene knockdown well, we are unable to capture late infection events such as virus release or infectivity of progeny virions. Hence our screening assay is limited in capturing factors that affect entry and post-entry replication events but not virus release. Although we did not identify apoptotic factors,

several autophagy involved cellular factors were identified in the primary and subsequent secondary screens (see Discussion Lines 787-810). This is possibly due to the involvement of autophagy components during intracellular replication prior to virus release/spread events.

Line 152: the authors do not comment on the fact that the relative infection ratio between control cells and silenced cell (no matter the gene) is greater in SK cells than in RD cells. Does it depend on the greater efficiency of transfection and silencing in the neuronal cell line? It will be important to demonstrate the silencing efficiency in both cell lines has by for example Western blotting or qRT-PCR and this at least for the 16 hits that were selected for further study.

We thank the reviewer for pointing out this discrepancy between results in RD and SK cells. We have since performed Western blot for 3 of the selected factors CDK6, RDX and AURKB in SK-N-SH and RD cells and found no significantly greater silencing efficiency in SK-N-SH when compared against that in RD cells (Fig S7a).

We believe that the more pronounced effects of knockdown on EV71 replication in SK-N-SH cells are due to the slower replication rate of EV71 in SK-N-SH cells compared to that in RD cells (see Fig S7b). All SK-N-SH data shown are taken at an endpoint of 24hpi while those in RD cells are at 12hpi. As can be seen from the growth kinetics, EV71 titer in RD peaks rapidly at 12hpi whereas that in SK-N-SH continues to climb gradually after 24hpi. We have included a discussion paragraph on this observation in Lines 642-656.

Fig.4a The figure is confusing. Some factors are not mentioned in the text and do not show complementary behavior compared to figure 3a (for instance, the overexpression of PMVK and CNOT3 do not increase the relative infection of EV-A71 even though their silencing decreases EV-A71 relative infection). By consequence, the finding on DTX1 lacking the RING finger is not informative in absence of the full-length DTX1 control. Overall the findings on DTX1 are too preliminary to be presented and given the density of results reported in the paper, this part can be eliminated without affecting the integrity and impact of the paper (see also comments above). Same remark for the paragraph 'disrupting the mevalonate pathway inhibits EV71 replication', even though the authors try to explain the counterintuitive effect of the overexpression of PMVK, no additional results are presented in this paragraph. Hence this section may be omitted though some brief mentioning of these findings in the discussion may be relevant.

All reviewers have suggested the removal of preliminary data and we agree to this consensus and removed all the preliminary results and speculative discussions involving DTX1, PMVK, CD1a and TGFB111 etc, and chose to focus on CDK6, AURKB, NGLY1 and VCP. We hope the reviewer will find a more succinct and focused manuscript in our revised version.

Minor comments

-Line 71: reference Coyne et al missing

Proper referencing has been included.

-Line 274: the authors take for granted that NGLY1 recruitment and glycanase activity are dependent on the PUB domain and the R401X mutation. This has to be stated in the text.

We have described the experimental and literature basis for our choice of mutations although we have not verified the NGLY1 activities of these mutants in our own setup. These explanations can be found in Lines 418-425.

-Line 407: `host transcription is necessary for enterovirus replication at early time point of infection` is an old finding (~70s). The best demonstration is the effect of actinomycin D in the context of poliovirus infection, which decrease viral yield at early times post-infection.

We thank the reviewer for pointing this out and have since removed the portion on host transcription from the Discussion section.

REVIEWERS' COMMENTS:

Reviewer #1 (Remarks to the Author):

The comments raised by this reviewer have been addressed by the authors. This manuscript lists up many new host factors supporting or inhibiting EV71 replication. Although the authors deleted some of the data, this reviewer still thinks that the discussion of the manuscript is too long and that some are just speculations and they can be deleted.

line 153; Fig. 3a should be Fig.2

line 177, 183, 188, 192; Fig 5 should be Fig 3

Reviewer #2 (Remarks to the Author):

The revised version of the manuscript by Kan Xing Wu and co-workers has addressed the majority of the criticisms raised by this reviewer, in particular those concerning the lack of focus of the initial version and the superficiality of the follow-up experiments. Although the phenotypes characterized for some of the hits are of low magnitude, they provide interesting and novel insights into the interplay between Enterovirus 71 and host cells.

The dataset itself constitutes a useful resource for researchers working in the field of host-pathogen interactions.

Reviewer #3 (Remarks to the Author):

The authors addressed the comments. They also shortened the Results section by removing the preliminary findings. Overall, I find the manuscript still overloaded and the Discussion still a bit too long.

Minor comment:

*There are mistakes in quoting the figures in the text. To be carefully revised

Line 148: Fig 2b and c and NOT Fig 3b and c

Line 153: fig 2a Not Fig 3a

Line 177: Fig 3a and d NOT fig 5a and d. Same for line 188, 192

-Line 194-205: This should be moved to the discussion.

REVIEWERS' COMMENTS:

Reviewer #1 (Remarks to the Author):

The comments raised by this reviewer have been addressed by the authors. This manuscript lists up many new host factors supporting or inhibiting EV71 replication. Although the authors deleted some of the data, this reviewer still thinks that the discussion of the manuscript is too long and that some are just speculations and they can be deleted.

line 153; Fig. 3a should be Fig.2

line 177, 183, 188, 192; Fig 5 should be Fig 3

Authors' reply

We thank the reviewer for the constructive comments and have removed the last paragraph of the discussion describing a proposed hypothesis explaining the propensity of EV71 infections in young children. We also appreciate the careful reading and correction of our mistakes in quoting figures in text. We have since thoroughly read through the manuscript to ensure correct figures are quoted.

Reviewer #2 (Remarks to the Author):

The revised version of the manuscript by Kan Xing Wu and co-workers has addressed the majority of the criticisms raised by this reviewer, in particular those concerning the lack of focus of the initial version and the superficiality of the follow-up experiments. Although the phenotypes characterized for some of the hits are of low magnitude, they provide interesting and novel insights into the interplay between Enterovirus 71 and host cells. The dataset itself constitutes a useful resource for researchers working in the field of host-pathogen interactions.

Authors' reply

We thank the reviewer for the comments given and constructive review in improving our manuscript.

Reviewer #3 (Remarks to the Author):

The authors addressed the comments. They also shortened the Results section by removing the preliminary findings. Overall, I find the manuscript still overloaded and the Discussion still a bit too long.

Minor comment:

*There are mistakes in quoting the figures in the text. To be carefully revised

Line 148: Fig 2b and c and NOT Fig 3b and c

Line 153: fig 2a Not Fig 3a

Line 177: Fig 3a and d NOT fig 5a and d. Same for line 188, 192

-Line 194-205: This should be moved to the discussion.

Authors' reply

We thank the reviewer and have removed the last paragraph of the discussion describing a proposed hypothesis explaining the propensity of EV71 infections in young children to reduce the length of Discussion. We have also taken up the advise of moving Line 194-205 to Discussion from the Results. Errors made in quoting figures have also been corrected.